

# Competing endogenous RNA network identifies mRNA biomarkers for overall survival of lung adenocarcinoma: two novel on-line precision medicine predictive tools

Jinsong Lin[*], Shubiao Lu[*], Zhijian Jiang[*], Chongjing Hu[*] and Zhiqiao Zhang[*]

Department of Internal Medicine, The Affiliated Chencun Hospital of Shunde Hospital, Southern Medical University, Shunde, Guangdong, china
[*] These authors contributed equally to this work.

## ABSTRACT

**Background**. Individual mortality risk predicted curve at the individual level can provide valuable information for directing individual treatment decision. The present study attempted to explore potential post-transcriptional biological regulatory mechanism related with overall survival of lung adenocarcinoma (LUAD) patients through competitive endogenous RNA (ceRNA) network and develop two precision medicine predictive tools for predicting the individual mortality risk curves for overall survival of LUAD patients.

**Methods**. Multivariable Cox regression analyses were performed to explore the potential prognostic indicators, which were used to construct a prognostic model for overall survival of LUAD patients. Time-dependent receiver operating characteristic (ROC) curves were used to assess the predictive performance of prognostic model.

**Results**. There were 494 LUAD patients in model cohort and 233 LUAD patients in validation cohort. Differentially expressed mRNAs, miRNAs, and lncRNAs were identified between LUAD tissues and normal tissues. A ceRNA regulatory network was constructed on previous differentially expressed mRNAs, miRNAs, and lncRNAs. Fourteen mRNA biomarkers were identified as independent risk factors by multivariate Cox regression and used to develop a prognostic model for overall survival of LUAD patients. The C-indexes of prognostic model in model group were 0.786 (95% CI [0.744–0.828]), 0.736 (95% CI [0.694–0.778]) and 0.766 (95% CI [0.724–0.808]) for one year, two year and three year overall survival respectively. Two precision medicine predicted tools were developed for predicting individual mortality risk curves for LUAD patients.

**Conclusion**. The current study explored potential post-transcriptional biological regulatory mechanism and prognostic biomarkers for overall survival of LUAD patients. Two on-line precision medicine predictive tools were helpful to predict the individual mortality risk predicted curves for overall survival of LUAD patients. Smart Cancer Survival Predictive System could be used at https://zhangzhiqiao2.shinyapps.io/Smart_cancer_predictive_system_9_LUAD_E1002/.

Corresponding author
Zhiqiao Zhang, sdgrxjbk@smu.edu.cn

## Background

Lung cancer is the most common cause of malignant tumours and tumour-related death (*Bray et al., 2018*). There were approximately 2.1 million newly diagnosed lung cancer patients and 1.8 million lung cancer deaths worldwide in 2018 (*Bray et al., 2018*). As the most prevalent type of non-small-cell lung cancer, lung adenocarcinoma (LUAD) is the leading cause of lung cancer-related mortality (*Bray et al., 2018*). The overall survival of patients with LUAD is extremely poor, with a 5-year overall survival (OS) rate of less than 20% (*Lin et al., 2016*). Although great progress has been made in cancer diagnosis and targeted therapy, the 5-year overall survival rate of patients with LUAD remains low (*Qi et al., 2016*). Therefore, establishing a reliable prognostic model for screening high-risk patients with poor overall survival is of great clinical significance for optimizing individualized treatments and improving the management of patients with cancer.

Several studies have reported potential molecular biological regulation mechanisms for different tumours (*Huang et al., 2018*; *Shi et al., 2018*; *Zeng et al., 2018*; *Zhong et al., 2018*). Salmena et al. proposed a posttranscriptional molecular biological regulation mechanism named competing endogenous RNA (ceRNA) (*Salmena et al., 2011*). Long noncoding RNAs (lncRNAs), as endogenous molecular sponges, can upregulate the expression of mRNA through competitive binding with miRNA response elements. (*Thomson & Dinger, 2016*). Some researchers have used a ceRNA regulatory network to explore the potential molecular biological regulation mechanism for the prognosis of patients with LUAD. (*Li et al., 2018*; *Sui et al., 2016*; *Wang et al., 2018*; *Wang et al., 2019*). *Li et al. (2018)* developed a prognostic signature (without external validation) for the prognosis of lung cancer by using 29 mRNAs and three lncRNAs . However, this prognostic model is too complex to calculate and be used for clinical application.

A nomogram is a graphical tool that can directly display the results, with the advantages of easy calculation and good interpretability (*Cheng, 2018*; *Song, Miao & Chen, 2018*). Based on the prognostic nomogram, our research team has further developed precision medicine predictive tools for providing individual mortality risk prediction curves for different cancers (*Cheng et al., 2019*; *Zhang et al., 2019b*; *Zhang et al., 2019c*). Therefore, the present study attempted to construct two precision medicine predictive tools for providing individualized mortality risk prediction curves for the OS of LUAD patients by using prognostic mRNA biomarkers through a ceRNA regulatory network.

## MATERIAL AND METHODS

### Protocol approval

Data were collected as described in our previous studies (*Zhang et al., 2019a*; *Zhang et al., 2019b*). The study data in the current study were downloaded from The Cancer Genome Atlas (TCGA) database (https://www.cancer.gov/about-nci/organization/ccg/research/structural-genomics/tcga), UCSC Xena database (https://xena.ucsc.edu/), cBioPortal

database (http://www.cbioportal.org/), and Gene Expression Omnibus (GEO) database (https://www.ncbi.nlm.nih.gov/geo/). The current study did not require ethics approval, as all study datasets were downloaded and analysed in accordance with the corresponding data policies of previous databases.

## The gene expression information of model cohort

Data were collected as described in our previous studies (*Zhang et al., 2020*; *Zhang et al., 2019b*). The current study downloaded the original RNA expression information from TCGA database (https://tcga-data.nci.nih.gov/docs/publications/tcga/) on the Illumina HiSeq 2000 RNA Sequencing platform. The RNA symbols were determined according to GENCODE Version 29 (https://www.gencodegenes.org/human/). The RNA expression dataset contained 8,848 lncRNAs and 21,138 mRNAs from 535 lung adenocarcinoma samples and 59 normal samples. The miRNA expression dataset was obtained from the UCSC Xena database (https://xena.ucsc.edu/). The miRNA expression dataset involved 1,881 miRNAs from 518 lung adenocarcinoma and 46 normal samples. According to the median of the original gene expression value, the original expression values were converted into "0" for lower expression and "1" for higher expression.

## Differentially expressed analyses

Differentially expressed analyses between tumour samples and normal samples were performed by the edgeR package with R software (version 3.5.2) as described in our previous studies (*Zhang et al., 2019b*; *Zhang et al., 2018*). For differentially expressed analyses, *a* false discovery rate (FDR) less than 0.05 and a ratio value of 1.5 between tumour tissues and normal tissues were defined as the thresholds.

## The clinical information of model cohort

Data were collected as described in our previous studies (*Zhang et al., 2019b*; *Zhang et al., 2018*). The clinical information was obtained from the cBioPortal database. Patients with LUAD with inadequate survival information or OS times less than one month (for living patients) were removed ($n = 19$). Therefore, 503 LUAD patients were identified with adequate OS information. There were 494 patients with adequate gene detection information and survival information when performing intersection between clinical datasets on the gene detection dataset.

## The corresponding information of validation cohort

Data were collected as described in our previous studies (*Zhang et al., 2019b*; *Zhang et al., 2018*). The current study downloaded GSE37745 and GSE50081 as external validation datasets from the GEO database. Gene expression information was detected on the GPL570 platform (Affymetrix Genome U133 Plus 2.0 Array). The verification group contained 233 LUAD patients and 22,850 RNA expression count values.

## Statistical analysis

The statistical analyses were carried out as described in our previous studies (*Zhang et al., 2020*; *Zhang et al., 2019b*): continuous variables are presented as the mean ± standard deviation (SD) or median (first percentile, third percentile) as appropriate. Continuous

variables were compared by *t*-test or Mann–Whitney U test as appropriate. Categorical variables were compared through the chi-squared test or Fisher's exact test as appropriate. Time-dependent receiver operating characteristic (ROC) curves were used to assess the predictive performance of prognostic models. Decision curve analysis (DCA) is an assessment method to compare the predictive performance of prognostic models (*Localio & Goodman, 2012*; *Vickers et al., 2008*; *Vickers & Elkin, 2006*). The statistical analyses were carried out using SPSS Statistics 19.0 (SPSS Inc., USA) and R software (version 3.5.2) with the following packages: "GOplot", "timeROC", "rms", "pROC", "survival", "clusterProfler" and "glmnet", as described in our previous studies (*Zhang et al., 2020*; *Zhang et al., 2019b*). A *P* value < 0.05 was the threshold value for statistical significance.

# RESULTS

## Study cohorts

In the model group (Doc S1), 182 patients (36.8%) out of 494 patients died during the follow-up period, while 127 patients (54.5%) out of 233 patients died in the verification group (Doc S2). The basic features of the two groups are compared in Table 1.

## Differentially expressed analyses

Based on the given threshold, 3,310 upregulated lncRNAs, 675 downregulated lncRNAs, 95 upregulated miRNAs, 30 downregulated miRNAs, 4,913 upregulated mRNAs, and 1,921 downregulated mRNAs were identified as differentially expressed RNAs between LUAD tissues and normal tissues. Based on previous differentially expressed mRNAs, 2,982 mRNAs were identified as prognostic mRNA indicators by univariate Cox analyses.

## Construction of a competing endogenous RNA network

To obtain the lncRNA-miRNA pairs, the miRcode database (http://www.mircode.org/) was searched according to the differentially expressed lncRNAs. To obtain the miRNA-mRNA pairs, TargetScan (http://www.targetscan.org/), miRDB (http://mirdb.org/), and miRTarBase (https://bio.tools/mirtarbase) were searched according to the previous lncRNA-targeted miRNAs. The miRNA-predicted mRNAs that could be searched in three databases were defined as miRNA-targeted mRNAs. Then, we determined the interaction between miRNA-targeted mRNAs and prognostic mRNAs to identify prognosis-associated miRNA-targeted mRNAs. Finally, twenty-two lncRNAs, twenty-nine miRNAs, and seventy-three mRNAs were used to build the ceRNA network for the overall survival of Patients with LUAD. The ceRNA regulatory network was depicted by Cytoscape software (Fig. 1).

## Functional enrichment analyses

To explore the biological functions of prognostic mRNAs, Gene Ontology (GO) and Kyoto Encyclopedia of Genes and Genomes (KEGG) analyses were performed on prognostic mRNAs in the ceRNA network. The bar plot and dot plot of prognostic mRNAs are depicted in Figs. 2A and 2B. GO terms and KEGG pathways of prognostic mRNAs are depicted in Figs. 3A and 3B.

**Table 1** The clinical features of lung adenocarcinoma patients in the model group and validation group.

|  | Model group ($n = 494$) | Validation group ($n = 233$) | *P* value |
|---|---|---|---|
| Death [n(%)] | 182(36.8) | 127(54.5) | <0.001 |
| Survival time for living patients(mean ± SD, month) | 22.2(15.6,37.6) | 69.1(59.3,88.0) | <0.001 |
| Survival time for dead patients (mean ± SD, month) | 20.3(9.8,36.8) | 25.3(12.4,48.4) | 0.006 |
| Age (mean ± SD, year) | 65.3 ± 10.0 | 66.1 ± 9.9 | 0.318 |
| Male [(n)%] | 229(46.4) | 111(47.6) | 0.746 |
| AJCC Stage (IV) | 25 | 4 | <0.001 |
| AJCC Stage (III) | 80 | 13 | |
| AJCC Stage (II) | 116 | 54 | |
| AJCC Stage (I) | 273 | 162 | |
| AJCC Stage (NA) | 0 | 0 | |
| AJCC PT (T1) | 229 | NA | NA |
| AJCC PT (T0) | 265 | NA | |
| AJCC PT (NA) | 0 | NA | |
| AJCC PN (N4) | 18 | NA | NA |
| AJCC PN (N3) | 44 | NA | |
| AJCC PN (N2) | 264 | NA | |
| AJCC PN (N1) | 165 | NA | |
| AJCC PN (N0) | 3 | NA | |
| AJCC PN (NA) | 0 | NA | |
| AJCC PM (M3) | 2 | NA | NA |
| AJCC PM (M2) | 69 | NA | |
| AJCC PM (M1) | 92 | NA | |
| AJCC PM (M0) | 302 | NA | |
| AJCC PM (NA) | 11 | NA | |

**Note**

Continuous variables were compared by *t*-test or Mann-Whitney *U* test as appropriate; Categorical variables were compared by chi-squared test or Fisher's exact test as appropriate.

NA, missing data; SD, standard deviation; AJCC, American Joint Committee on Cancer.

## Development of a prognostic nomogram

The previous prognostic mRNAs were used to construct a prognostic model for overall survival. The information on these mRNAs is summarized in Table 2. The risk score of the prognostic model was calculated by using the following formula: risk score = $(-0.7305*DNAJC27) + (0.4776*NPAS2) + (0.3941*PHKA1) + (-0.5537*CDADC1) + (0.4792*PTGFRN) + (0.3980*DDIT4) + (-0.4133*SCAMP5) + (-0.4367*SPRY2) + (0.3850*CSE1L) + (-0.3940*ELAVL4) + (0.3293*TRIM29) + (0.3685*LPGAT1) + (-0.3279*TRPC3) + (0.3257*DCBLD2)$. A prognostic nomogram chart is presented in Fig. 4. Kaplan–Meier survival curves (Fig. S1 ) demonstrated that these 14 prognostic mRNAs were significantly correlated with OS ($P < 0.05$). The results of differential expression analysis of 14 enrolled mRNAs are listed in Table 3.

## Predictive performance of the prognostic model

Based on the median value, LUAD patients in the model group could be classified into a high-risk subgroup and a low-risk subgroup. The overall survival rate (Fig. 5A) in the

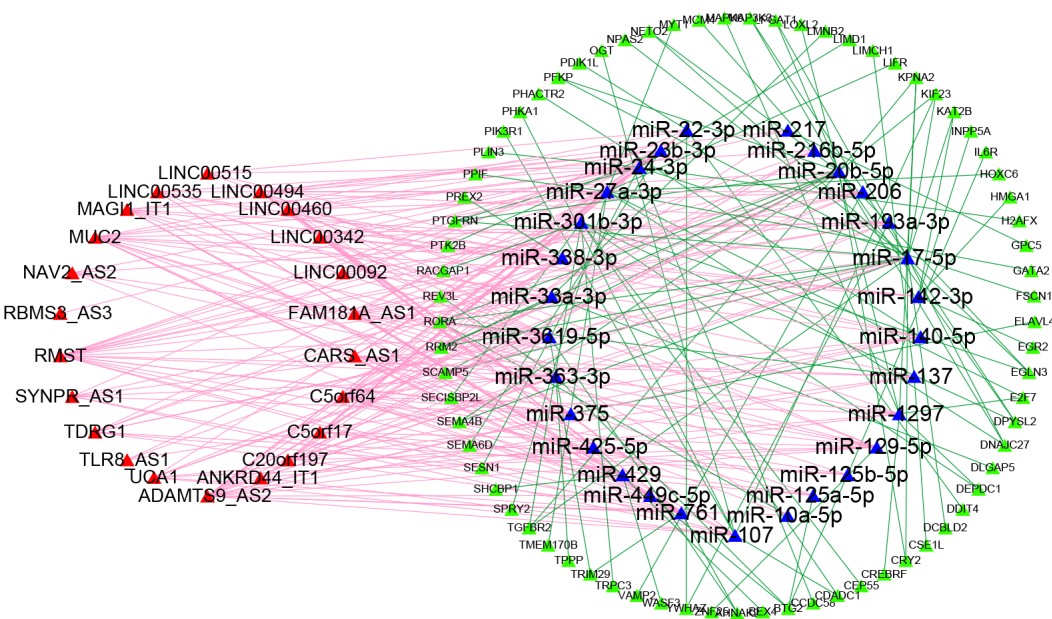

**Figure 1** Competitive endogenous RNA network chart: the red triangles represent 22 lncRNAs; the blue triangles represent 29 miRNAs; the green circles represent 73 mRNAs.

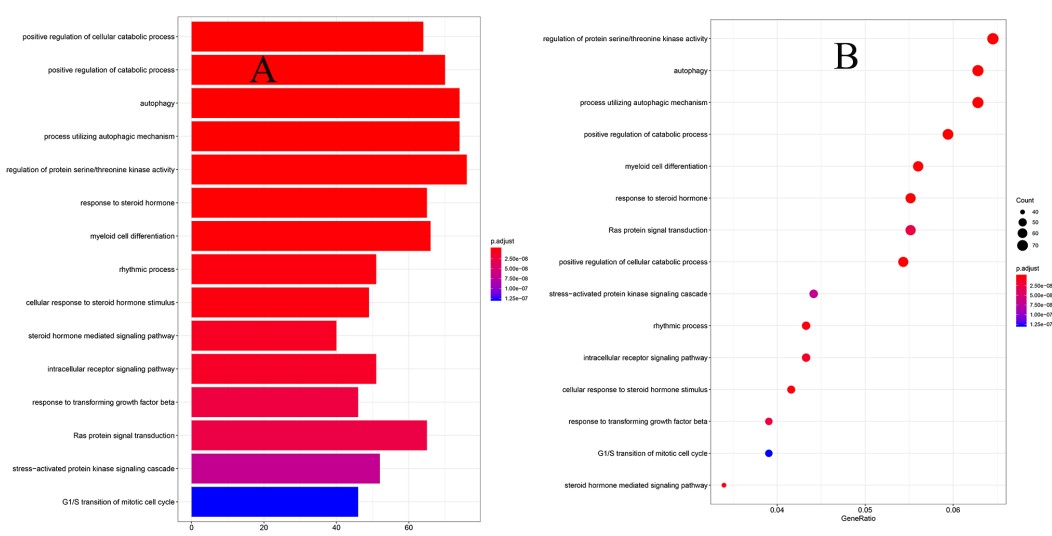

**Figure 2** (A) The barplot of GO terms for prognostic mRNAs; (B) the dotplot of GO terms for prognostic mRNAs.

high-risk group was significantly poorer than that in the low-risk group ($P < 0.001$). Harrell's concordance indexes (C-indexes) of the prognostic signature for overall survival in the model group were 0.786 (95% CI [0.744–0.828]), 0.736 (95% CI [0.694–0.778]), and 0.766 (95% CI [0.724–0.808]) for 1-year, 2-year, and 3-year OS, respectively (Fig. 5B). The scatter plot and the interaction distribution scatter plot are presented in Fig. 5C and
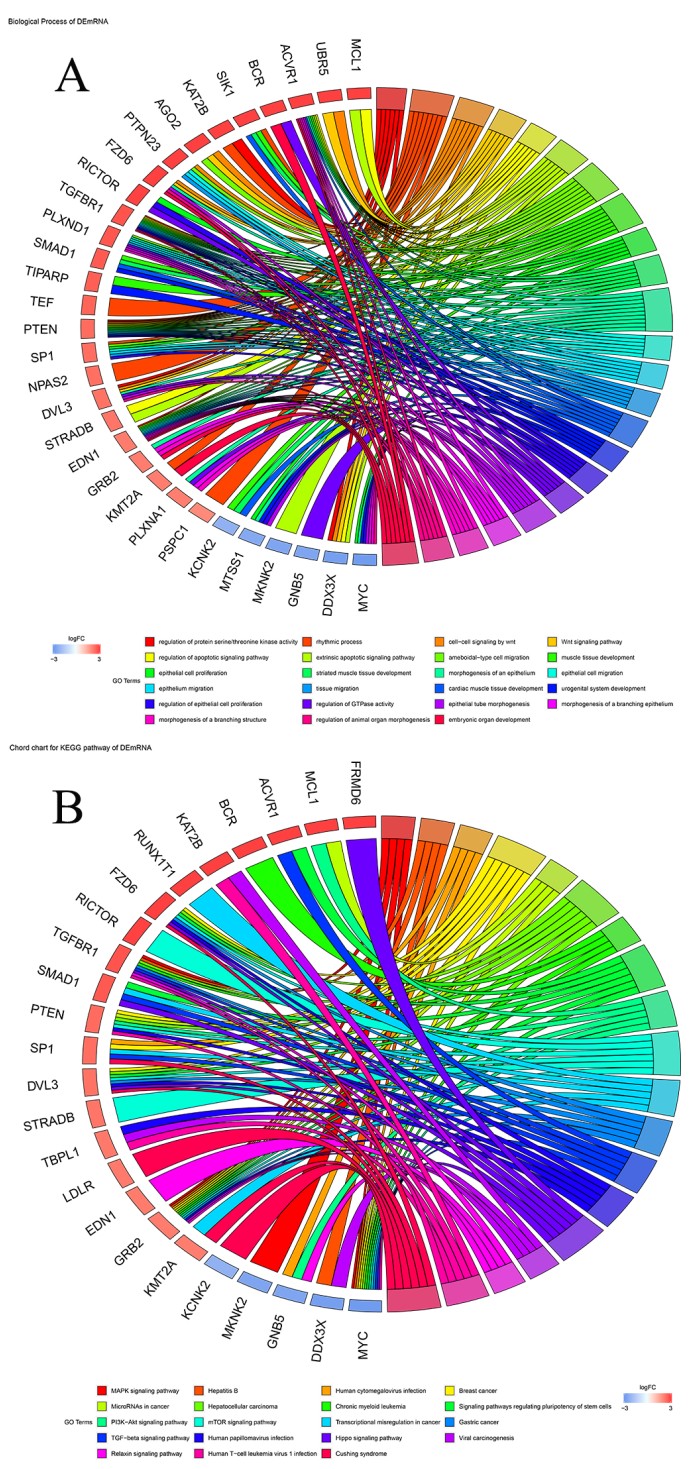

**Figure 3** (A) Chord chart of GO terms for prognostic mRNAs; (B) KEGG pathways for prognostic mRNAs.

**Table 2  The model information of prognostic mRNAs in univariate and multivariable Cox regression analyses.**

| Gene | Univariate analysis | | | Multivariate analysis | | | |
|---|---|---|---|---|---|---|---|
| | HR | 95% CI | *P*-value | coefficient | HR | 95% CI | *P*-value |
| DNAJC27 (High/Low) | 0.594 | 0.441–0.800 | <0.001 | −0.731 | 0.482 | 0.336–0.691 | <0.001 |
| NPAS2 (High/Low) | 1.613 | 1.202-2.164 | <0.001 | 0.478 | 1.612 | 1.166–2.229 | 0.004 |
| PHKA1 (High/Low) | 1.367 | 1.020–1.832 | 0.036 | 0.394 | 1.483 | 1.032–2.131 | 0.033 |
| CDADC1 (High/Low) | 0.699 | 0.521–0.936 | 0.016 | −0.554 | 0.575 | 0.394–0.839 | 0.004 |
| PTGFRN (High/Low) | 1.356 | 1.012–1.818 | 0.042 | 0.479 | 1.615 | 1.120–2.329 | 0.010 |
| DDIT4 (High/Low) | 1.634 | 1.217–2.193 | <0.001 | 0.398 | 1.489 | 1.086–2.042 | 0.013 |
| SCAMP5 (High/Low) | 0.737 | 0.550–0.988 | 0.041 | −0.413 | 0.662 | 0.474–0.924 | 0.015 |
| SPRY2 (High/Low) | 0.714 | 0.533–0.957 | 0.024 | −0.437 | 0.646 | 0.455–0.918 | 0.015 |
| CSE1L (High/Low) | 1.342 | 1.001–1.798 | 0.049 | 0.385 | 1.470 | 1.024–2.110 | 0.037 |
| ELAVL4 (High/Low) | 0.707 | 0.527–0.950 | 0.021 | −0.394 | 0.674 | 0.496–0.918 | 0.012 |
| TRIM29 (High/Low) | 1.391 | 1.037–1.864 | 0.028 | 0.329 | 1.390 | 1.010–1.912 | 0.043 |
| LPGAT1 (High/Low) | 1.410 | 1.052–1.890 | 0.022 | 0.369 | 1.446 | 1.017–2.054 | 0.040 |
| TRPC3 (High/Low) | 0.681 | 0.506–0.915 | 0.011 | −0.328 | 0.720 | 0.525–0.988 | 0.042 |
| DCBLD2 (High/Low) | 1.346 | 1.005–1.802 | 0.046 | 0.326 | 1.385 | 1.001–1.915 | 0.049 |

**Notes.**

HR, hazard ratio; CI, confidence interval.
The medians of mRNA expression values were used as cut-off values to stratify mRNA expression values into high expression group (as value 1) and low expression group (as value 0).

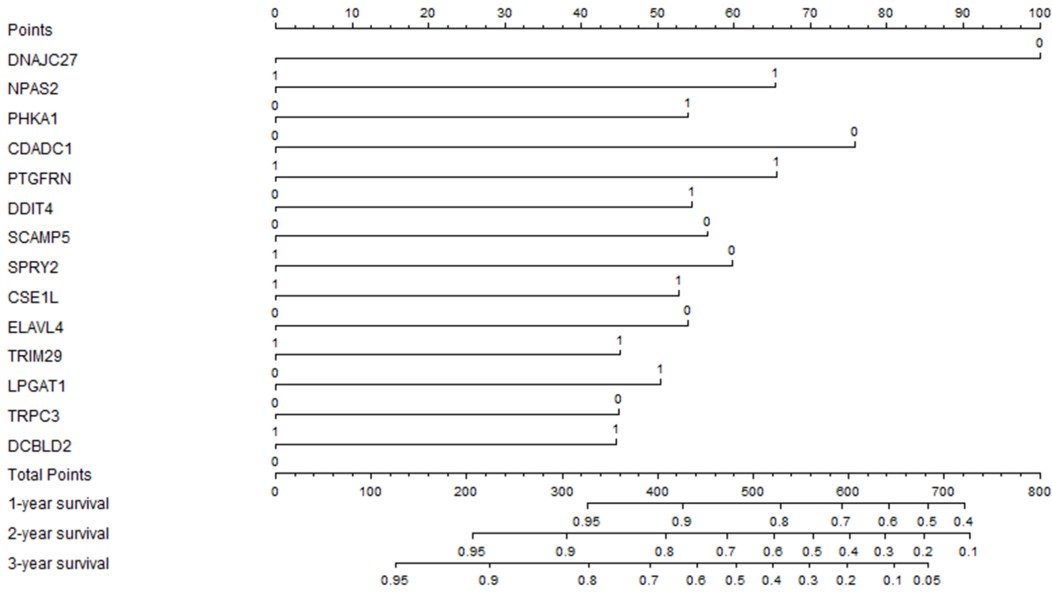

**Figure 4  The prognostic nomogram for overall survival.**

Fig. 5D. Calibration curves are depicted in Fig. S2A for 1 year, Fig. S2B for 2 years, and Fig. S2C for 3-year overall survival.
**Table 3    Results of differential expression analysis of the enrolled mRNAs. .**

| Symbol | F | *P* value | FC | logCPM | FDR |
|---|---|---|---|---|---|
| PHKA1 | 207.115 | <0.001 | 3.252 | 4.717 | <0.001 |
| CDADC1 | 125.913 | <0.001 | 0.666 | 2.854 | <0.001 |
| LPGAT1 | 116.687 | <0.001 | 2.218 | 6.843 | <0.001 |
| PTGFRN | 104.576 | <0.001 | 2.304 | 6.604 | <0.001 |
| DNAJC27 | 95.382 | <0.001 | 0.621 | 2.815 | <0.001 |
| SCAMP5 | 66.600 | <0.001 | 2.384 | 4.159 | <0.001 |
| CSE1L | 63.349 | <0.001 | 1.694 | 6.671 | <0.001 |
| NPAS2 | 52.396 | <0.001 | 2.416 | 4.793 | <0.001 |
| TRPC3 | 44.057 | <0.001 | 0.429 | −0.488 | <0.001 |
| SPRY2 | 42.334 | <0.001 | 0.629 | 5.171 | <0.001 |
| ELAVL4 | 41.979 | <0.001 | 3.749 | −0.076 | <0.001 |
| DDIT4 | 34.794 | <0.001 | 2.249 | 7.135 | <0.001 |
| DCBLD2 | 30.157 | <0.001 | 2.262 | 7.216 | <0.001 |
| TRIM29 | 7.324 | 0.007 | 2.108 | 5.400 | 0.010 |

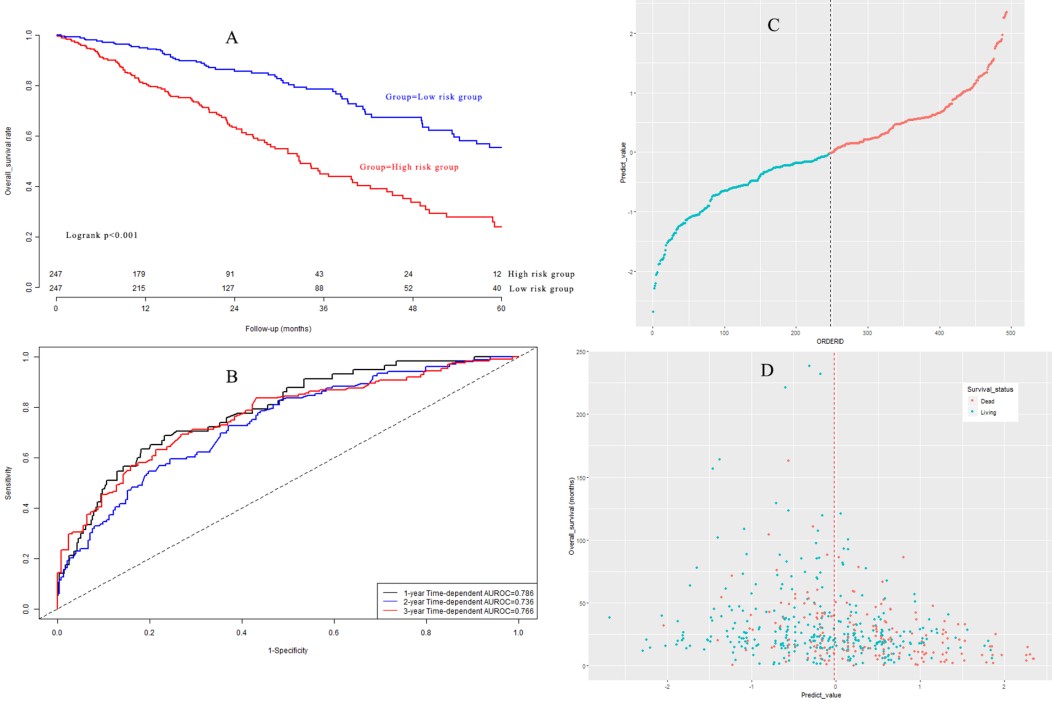

**Figure 5**   (A) Survival curves in model group; (B) time-dependent receiver operating characteristic curves in model group; (C) the distribution of prognostic model score in model group; (D) the overall survival status and overall survival time in model group.

## External validation of the prognostic model

Kaplan–Meier analysis (Fig. 6A) indicated that there was a significant difference in OS between the low-risk group and the high-risk group in the validation dataset ($P < 0.001$).

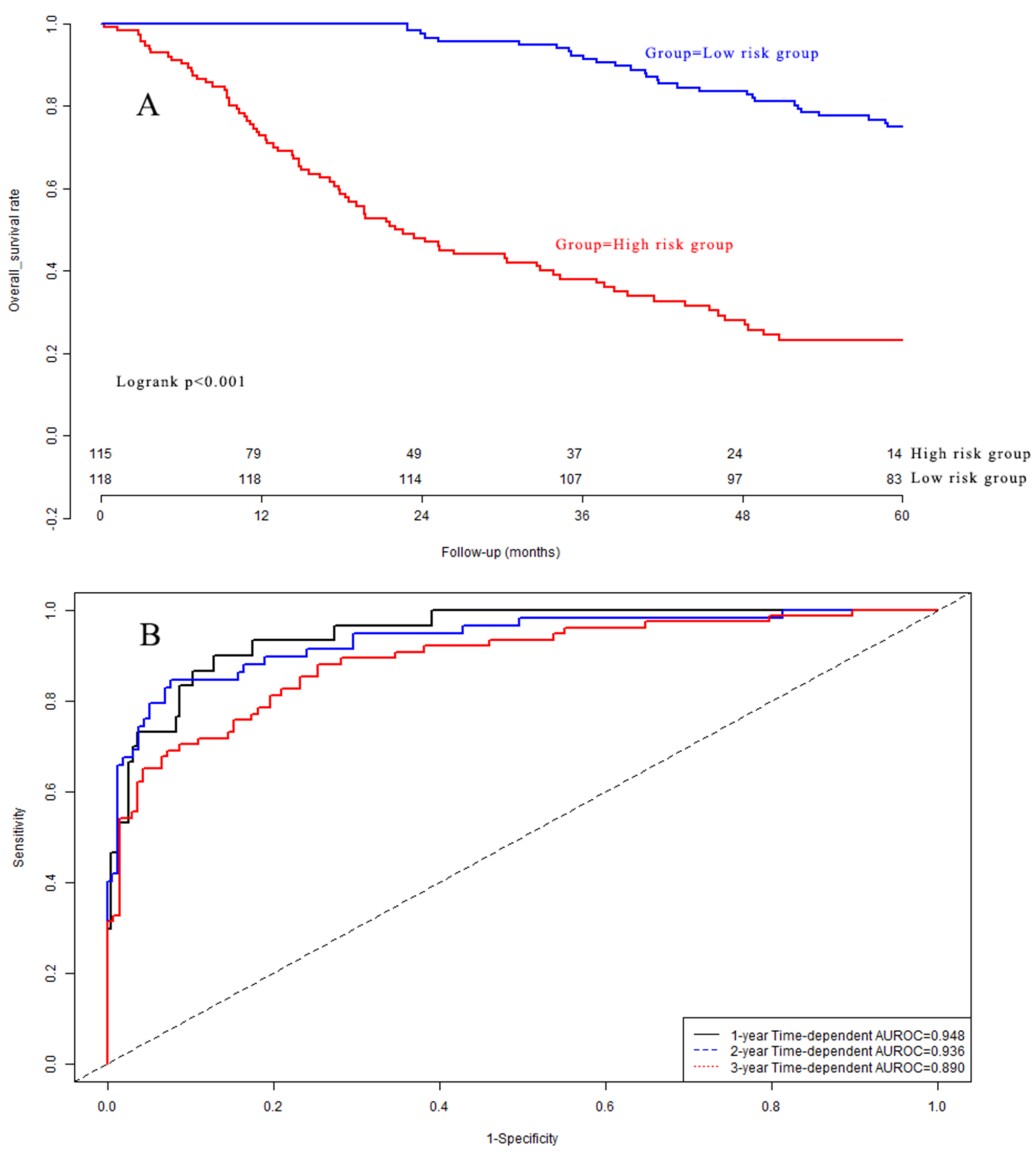

**Figure 6** (A) Survival curves in validation group; (B) time-dependent receiver operating characteristic curves in validation group.

The C-indexes of the prognostic model for overall survival in the validation dataset were 0.948 (95% CI [0.912–0.984]), 0.936 (95% CI [0.900–0.972]), and 0.890 (95% CI [0.854–0.926]) for 1-year, 2-year, and 3-year overall survival, respectively (Fig. 6B). Calibration curves are depicted in Fig. S3A for 1 year, Fig. S3B for 2 years, and Fig. S3C for 3-year overall survival.

## Independence assessment of the prognostic model

Table 4 shows that the prognostic model was an independent influencing factor for OS after adjustment for confounding effects in the model dataset. In the validation dataset, multivariate Cox regression analyses demonstrated that the prognostic model was an

**Table 4  Univariate and multivariable Cox regression analyses for independence assessment of prognostic model.**

| | Univariate analysis | | | Multivariate analysis | | | |
|---|---|---|---|---|---|---|---|
| | HR | 95% CI | *P*-value | coefficient | HR | 95% CI | *P*-value |
| Model group (*n* = 494) | | | | | | | |
| Age(≥60/<60 years) | 1.003 | 0.722–1.392 | 0.987 | 0.078 | 1.082 | 0.776–1.507 | 0.643 |
| Gender (Male/Female) | 1.048 | 0.783–1.403 | 0.754 | 0.077 | 1.080 | 0.805–1.449 | 0.609 |
| AJCC stage (IV,III/II,I) | 2.599 | 1.906–3.545 | <0.001 | 0.800 | 2.225 | 1.622–3.051 | <0.001 |
| Prognostic model (High/Low) | 2.705 | 1.986–3.684 | <0.001 | 0.895 | 2.447 | 1.786–3.352 | <0.001 |
| Validation group(*n* = 233) | | | | | | | |
| Age(≥60/<60 years) | 0.876 | 0.595–1.289 | 0.501 | −0.271 | 0.762 | 0.512–1.136 | 0.183 |
| Gender (Male/Female) | 1.308 | 0.921–1.857 | 0.134 | 0.268 | 1.308 | 0.911–1.877 | 0.146 |
| AJCC stage (IV,III/II,I) | 0.988 | 0.517–1.888 | 0.974 | 0.354 | 1.424 | 0.738–2.750 | 0.292 |
| Prognostic model (High/Low) | 5.109 | 3.428–7.348 | <0.001 | 1.651 | 5.214 | 3.531–7.701 | <0.001 |

**Notes.**
AJCC, the American Joint Committee on Cancer; HR, hazard ratio; CI, confidence interval.
The median of Prognostic model scores was used as the cut-off value to stratify gastric cancer patients into high risk group and low risk group.

independent influencing factor for OS. DCA is shown in Fig. S4A for 1-year OS, Fig. S4B for 2-year OS, and Fig. S4C for 3-year OS. The clinical impact curve is presented in Fig. S4D.

## Smart Cancer Survival Predictive System

A precision medicine predictive tool named the Smart Cancer Survival Predictive System was developed for predicting the prognosis of LUAD patients. The Smart Cancer Survival Predictive System (Fig. 7) is available at https://zhangzhiqiao2.shinyapps.io/Smart_cancer_predictive_system_9_LUAD_E1002/.
The Smart Cancer Survival Predictive System could predict full-time mortality risk prediction curves for one specific patient (Fig. 7A). Figure 7B, demonstrates the mortality rate predicted percentage and 95% confidence interval at different user-defined time points.

## Gene Survival Analysis Screen System

A second precision medicine predictive tool named the Gene Survival Analysis Screen System (Fig. 8) was developed to explore the survival features of prognostic mRNAs. The Gene Survival Analysis Screen System is available at https://zhangzhiqiao7.shinyapps.io/Gene_Survival_Analysis_Screening_System_9_LUAD_E1002/. Figure 8A depicts and compares the survival curves between two defined subgroups. Figure 8B displays the results of univariate survival analysis for selected variables.

## DISCUSSION

The current ceRNA regulatory network depicts lncRNA-miRNA-mRNA regulatory pathways, which are helpful for understanding the biological regulatory mechanisms of OS in LUAD. The current study constructed and verified a fourteen-mRNA prognostic nomogram for OS. This fourteen-mRNA prognostic nomogram was suitable to screen LUAD patients with poor OS. Based on this fourteen-mRNA prognostic nomogram, we

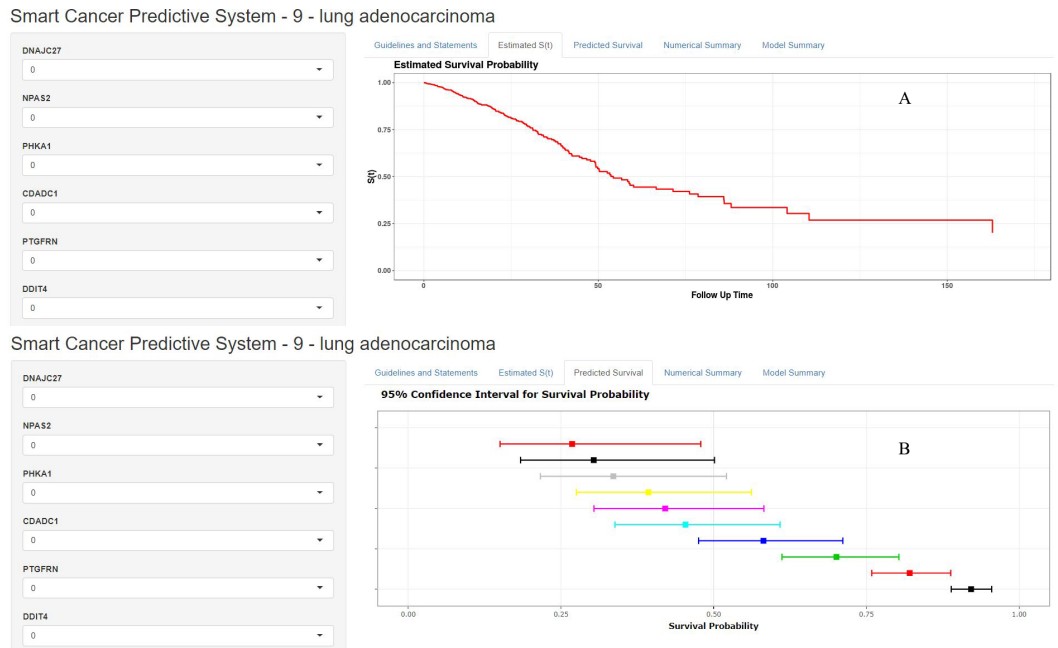

**Figure 7** Home page of Smart Cancer Survival Predictive System: (A) individual mortality risk predictive curve display page; (B) different time-point individual mortality risk prediction display page.

developed an online precision medicine predictive tool named the Smart Cancer Survival Predictive System, which can generate full-time mortality risk prediction curves for one specific patient at the individual level.

Several prognostic models have been built for predicting the prognosis of lung cancer patients (*Xie & Xie, 2019*; *Yan et al., 2018*; *Zuo et al., 2019*). However, these previous prognostic models could only provide prognostic information for a special subgroup at the group level and failed to provide individual mortality risk prediction at the individual level. Our Smart Cancer Survival Predictive System was superior to the previous prognostic models for its special predictive ability in predicting individual mortality risk curves at the individual level. Meanwhile, the Smart Cancer Survival Predictive System could further provide the mortality rate predicted percentage and 95% confidence interval at different user-defined time points. These special predictive functions in smart cancer survival predictive systems are of great significance for improving individual treatment decisions.

NPAS2 was associated with a favourable prognosis of LUAD patients (*Qiu et al., 2019*). CDADC1 was found to be associated with survival in bladder cancer (*Shivakumar et al., 2017*). DDIT4 promotes gastric cancer proliferation and tumorigenesis (*Du et al., 2018*). The knockdown or overexpression of SPRY2 promoted or suppressed the proliferation of prostate cancer cells (*Gao et al., 2018*). CSE1L was correlated with overall survival in patients with hepatocellular carcinoma (*Zhang et al., 2019c*). Ectopic expression of TRIM29 potentially contributes to metastasis and poor prognosis in patients with osteosarcoma (*Zeng et al., 2017*). Increased activity of TRPC3 channels is necessary for the development of ovarian cancer (*Yang et al., 2009*). DCBLD2 correlated with the overall survival in

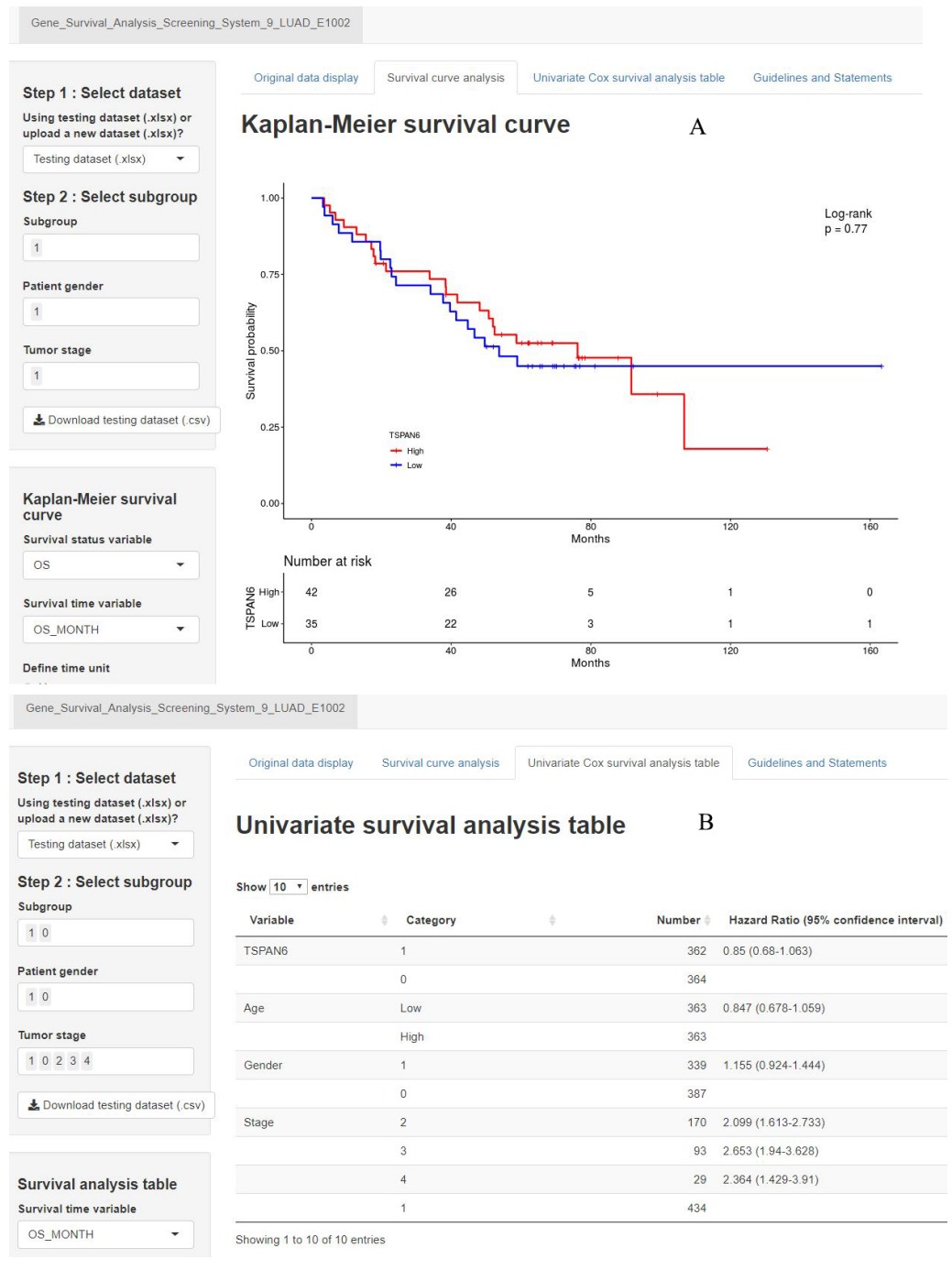

**Figure 8** Home page of Gene Survival Analysis Screen System: (A) survival curve plot display page; (B) univariate analysis table display page.

pancreatic ductal adenocarcinoma (*Raman et al., 2018*). The results of the current study were in good agreement with those of previous studies.

There are four main advantages of the current study: First, all operations are computed automatically in the background without manual calculation, which is convenient for use; Second, our tool can provide full-time individual mortality risk prediction through an individualized survival curve for a particular patient; Third, our tool can provide individual mortality risk prediction percentages and 95% CIs at specific time points (such as 12, 24, 36, 48, 60, and 72 months) through line graphs and tables; Fourth, the Gene Survival Analysis Screen System allows users to define different subgroups by themselves. Meanwhile, users are free to download, upload, and select the dataset for gene survival analysis. To the best of our knowledge, this online full-time individualized risk calculator is the first to provide full-time individual mortality risk prediction through a web calculator based on gene expression data for patients with lung cancer.

The shortcomings of our study were as follows: First, the genetic detection platforms of the model group and validation group were different and need to be taken into account when interpreting the results of the current study; Second, the performance of the fourteen-mRNA prognostic nomogram in the verification group was better than that in the model group, which could be explained by the longer follow-up period and higher mortality in the verification group. The median survival times were 20.3 and 25.3 months ($P = 0.006$) for patients who died in the model group and validation group, respectively, whereas the median survival times were 22.2 and 69.1 months ($P < 0.001$) for living patients in the model group and validation group, respectively; Third, some important prognostic factors, such as surgical procedures and adjuvant therapies, were not included in the survival analysis. It is necessary to carry out multicentre, prospective, and large-sample clinical research to further study the clinical application value of fourteen-mRNA prognostic nomograms in different populations.

## CONCLUSIONS

In conclusion, the current study explored potential posttranscriptional biological regulatory mechanisms and prognostic biomarkers for the overall survival of LUAD patients. Two online precision medicine predictive tools were developed and are helpful to predict the individual mortality risk prediction curves for the overall survival of LUAD patients. The Smart Cancer Survival Predictive System can be used at https://zhangzhiqiao2.shinyapps.io/Smart_cancer_predictive_system_9_LUAD_E1002/.

### Abbreviations

| | |
|---|---|
| **LUAD** | lung adenocarcinoma |
| **TCGA** | The Cancer Genome Atlas |
| **GEO** | Gene Expression Omnibus |
| **ROC** | receiver operating characteristic |
| **OS** | overall survival |
| **lncRNA** | long noncoding RNA |
| **miRNA** | microRNA |
| **mRNA** | messenger RNA |
| **HR** | hazard ratio |

| CI | confidence interval |
| AJCC | American Joint Committee on Cancer |
| SD | standard deviation |
| DCA | decision curve analysis |
| ceRNA | competitive endogenous RNA |

## ACKNOWLEDGEMENTS

We would like to express sincere thanks to Dr. Gary S Collins (University of Oxford), Dr Manali Rupji (Emory University), and Mrs Qingmei Liu for inspirations, suggestions, and assistance in the development of our precision medicine tools.

### Funding

This work was supported by the Guangdong Provincial Health Department (A2016450 and B2018237). The funders had no role in study design, data collection and analysis, decision to publish, or preparation of the manuscript.

### Grant Disclosures

The following grant information was disclosed by the authors:
Guangdong Provincial Health Department: A2016450, B2018237.

### Competing Interests

The authors declare there are no competing interests.

### Author Contributions

- Jinsong Lin, Shubiao Lu, Zhijian Jiang, Chongjing Hu and Zhiqiao Zhang conceived and designed the experiments, performed the experiments, analyzed the data, prepared figures and/or tables, authored or reviewed drafts of the paper, and approved the final draft.

### Human Ethics

The following information was supplied relating to ethical approvals (i.e., approving body and any reference numbers):

The current study did not require ethics approval as all study datasets were downloaded and analyzed in accordance with the corresponding data policies of previous databases.

### Data Availability

The study data is available in the Supplemental Files. The study data in the current study were downloaded from The Cancer Genome Atlas (TCGA) database (https://www.cancer.gov/about-nci/organization/ccg/research/structural-genomics/tcga), UCSC Xena database (https://xena.ucsc.edu/), cBioPortal database (http://www.cbioportal.org/), and the Gene Expression Omnibus (GEO) database (https://www.ncbi.nlm.nih.gov/geo/).

## Supplemental Information

Supplemental information for this article can be found online at http://dx.doi.org/10.7717/peerj.11412#supplemental-information.

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
