# Peer review of "Competing endogenous RNA network identifies mRNA biomarkers for overall survival of lung adenocarcinoma: two novel on-line precision medicine predictive tools"

_PeerJ, doi:10.7717/peerj.11412_

## Round 0.1 · original submission · Major Revisions

Dear Dr. Zhang,

Thank you for your submission to PeerJ.

Based on the reviewers' comments and my own opinion, the article requires MAJOR REVISION. As mentioned by Reviewer-1 and Reviewer-2, the main concerns are related to the cancer datasets used and the method applied. Please provide the accession numbers and other details of the lung adenocarcinoma datasets used in this research (its mentioned Lung adenocarcinoma in text, but the datasets used seems to be of Gastric cancer as per the tables). Also you need to apply an advanced method for the analysis. Further, the flow and writing of the article also needs to be improved.

Please go through each review comment and respond accordingly while you submit your revised article.

Thank you,
Kind regards,
Debmalya Barh, PhD

Reviewer 1 ·

Basic reporting

The article is well-written and the overall presentation is acceptable. Literature references are appropriate as well.

Experimental design

The authors explored potential post-transcriptional biological regulatory mechanism and prognostic biomarkers for overall survival of LUAD patients. Standard analysis tools were used and the study lacks methodological novelty.

The dataset was downloaded from The Cancer Genome Atlas (TCGA) database, UCSC Xena database, cBioPortal database, and the Gene Expression Omnibus (GEO) database.
This is a comprehensive dataset and adds to the strength of the paper.

The method used to identify biomarker/prognostic RNAs did not use more recent methodologies. Check the following paper for patient specific expression profiles:
Menche, J., Guney, E., Sharma, A. et al. Integrating personalized gene expression profiles into predictive disease-associated gene pools. npj Syst Biol Appl 3, 10 (2017). https://doi.org/10.1038/s41540-017-0009-0
I would recommend the authors to use this methodology for identifying the RNA sets of interest.

Validity of the findings

The authors mentioned:
According to the median of the original gene expression value, the original expression values were converted into “0” for lower expression and “1” for higher expression.
-- Did you use log fold change? Was any distribution used for this step or simply a median threshold was used? There are better methods in place already for this step and the rigor is not clear.

Authors mentioned:
Finally, twenty-two lncRNAs, twenty-nine miRNAs, and seventy-three mRNAs were used to build the ceRNA network for overall survival of LUAD patients.
-- Are these the only differentially expressed RNAs? The regulatory network was created based on the interaction mining from the three databases mentioned. But these are not based on only validated interactions. The authors need to correlate the regulatory network with the expression values seen in their dataset using reverse engineering algorithms.

There were several prognostic models built for predicting the prognosis for lung cancer patients (Xie & Xie 2019; Yan et al. 2018; Zuo et al. 2019).
-- How are the individual biomarkers compare with the group level results form these prior works? For validation, the authors show the relevance of the identified genes in different cancer scenarios; however, this discussion should be better focused for LUAD only.

Since the genetic detection platforms were different, some adjustments are needed to make these predictions more realistic.

Additional comments

Please see the comments mentioned above.

Reviewer 2 ·

Basic reporting

In this present study, authors developed prognostic models for overall survival of lung adenocarcinoma patients (LUAD). The major issue with this study is that authors used gastric cancer data (Table 1) and used this as LUAD patient data for all downstream analysis. Authors need to explicitly explain this discrepancy.
Other comments:
1) It is not clear how authors computed differential expression analysis. Please elaborate in the method section. The described cut off is 1.5 between tumor vs normal tissues. This is a very low bar and authors should include stringent cut off as well as include FDRs.
2) The figure legends need improvement. All legends should have enough description for a reader to understand the figure without having to refer back to the main text of the manuscript.
3) Please improve the resolution of figures.

Experimental design

no comment

Validity of the findings

no comment

Additional comments

no comment

Reviewer 3 ·

Basic reporting

The study explored potential post-transcriptional biological regulatory mechanism and prognostic biomarkers for overall survival of LUAD patients . The article is written well and unambiguous, technically correctly throughout. It includes sufficient introduction and background to demonstrate how the work fits into the broader field of knowledge. The structure of the article conform to an acceptable format of standard sections.

Experimental design

The experimental design within aims and scope of the journal. Research question well defined, relevant and meaningful. Methods described within sufficient detail and information to replicate.

Validity of the findings

Conclusions are well stated, linked to original research question and limited supporting results.

Additional comments

In this manuscript, the authors identified some novel differentially expressed RNAs and miRNAs between LUAD tumor tissues and noncancerous tissues, and they established a ceRNA network based on the differentially expressed genes. Two biomarkers candidates were identified from the ceRNA network. The topic is important, it may helpful in studying the pathomechanism of LUAD. English language and style are minor spell check required.

---

## Round 0.2 · Minor Revisions

Dear Dr. Zhang,

Your revised article is now reviewed and can be accepted. However, professional English Language editing is required before it is finally accepted. Please submit the article after Language editing for its official acceptance and further processing.

Thank you,
Best Regards,
Debmalya Barh.

Reviewer 1 ·

Basic reporting

This is fine.

Experimental design

The authors have properly addressed all my comments.

Validity of the findings

This is fine too.

Additional comments

A professional English check is needed before final publication.

---

## Round 0.3 · accepted · Accept

Dear Dr. Zhang,

Thank you for your submission to PeerJ.

I am glad you accept this revised and Language edited version.

Thank you,
Best Regards
Debmalya Barh, PhD

[